# Evaluation of Nationwide Oral Mucosal Screening Program for Oral Cancer Mortality among Men in Taiwan

**DOI:** 10.3390/ijerph192114329

**Published:** 2022-11-02

**Authors:** Shih-Yung Su

**Affiliations:** Master Program in Statistics, National Taiwan University, Taipei 10617, Taiwan; shihyungsu@ntu.edu.tw; Fax: +886-2-33661996

**Keywords:** oral cancer, oral mucosal screening, interrupted time-series analysis, risk fraction

## Abstract

The nationwide oral cancer screening program was launched for high-risk people (tobacco smokers or betel-nut chewers) in 1999 in Taiwan, but no study has taken the prevalence of tobacco smoking and betel-nut chewing into account for evaluating the impact of the screening program on oral cancer mortality. This study incorporated the risk fraction method with interrupted time-series analysis to evaluate the impact of the nationwide oral mucosal screening program among men in Taiwan. This study estimated the expected oral cancer mortality trend if the screening program had not launched in 1999, which revealed that the increasing oral cancer mortality trend would level off after 2009 due to the declining prevalence of tobacco smoking and betel-nut chewing. In 2000–2007, the percentage changes between the observed (implementation of the screening program) and expected (if the screening program had not launched) oral cancer mortality rate was not statistically significant for each age group. In 2008–2020, the significant percentage changes were −178% (99% CIs: −140.8 to −215.2), −75.4% (−59.4 to −91.4), −33.7% (−24.7 to −42.7), −18.8% (−12.0 to −25.6), and −15.3% (−9.5 to −21.2) for age groups of 30–34, 35–39, 40–44, 45–49, and 50–54, respectively. In addition to its influence on tobacco smoking and betel-nut chewing, the oral mucosal screening program was associated with the reduction of oral cancer mortality among men in Taiwan.

## 1. Introduction

Incidence and mortality of oral cancer (comprising lip, oral cavity, oropharynx, and hypopharynx cancers) are higher in the Asia-Pacific region and India [1]. The consumption of alcohol, tobacco, betel nuts, and smokeless tobacco products are the risk factors for oral cancer and also a major public health concern in the high-incidence region of oral cancer [2,3,4,5]. In the Asia-Pacific region, oral cancer was the fourth most common cancer in 2018 and also the fourth leading cause of cancer death in 2020 among men in Taiwan. For Taiwanese women, oral cancer is rare, and the age-standardized oral cancer incidence and mortality in men are approximately 10-fold higher than those in women [6,7]. In 1999, a nationwide oral cancer screening program was implemented in Taiwan. A selective screening and free oral mucosal examination every two years are performed for people older than 30 years old that have high-risk behaviors of tobacco smoking and betel-nut chewing. The details of the screening program are shown in Appendix B.

Understanding the impact of the national screening program on the morbidity of the target disease in the population could give important feedback to modify or improve the screening policy and strategy. However, it is difficult to evaluate the impact of the oral mucosal screening program because of the selective screening strategy which involves tobacco smoking and betel-nut chewing in the population. In other words, the expected screening effect (beneficial role) would be counteracted if the prevalence of smoking and chewing increased and would be ambiguous if the prevalence decreased. Previous studies using a database collected by the nationwide oral mucosal screening program in Taiwan evaluated the cost-effectiveness, reliability and validity, histopathology, and prognosis for oral cancer incidence and survival among eligible people who voluntarily receive screening and treatment [8,9,10]. However, these studies were screening-based or uncontrolled designs and cannot clarify the screening effect on population health. The previous study also indicated a leveling-off trend in oral cancer mortality in Taiwan since 2004, [11] but it could not clarify that this change is due to the screening program or the decreasing prevalence of tobacco smoking and betel-nut chewing. The risk factors (tobacco smoking and betel-nut chewing) are strongly associated with oral cancer mortality, but there is no study that takes risk factors into account for evaluating the impact of the screening program on population health.

The interrupted time-series (ITS) of quasi-experimental study design is commonly used to estimate the impact of health policy intervention on disease morbidity and burden through the temporal change of a disease trend before and after an intervention [12,13,14,15,16]. This study used the risk fraction method and the ITS analysis and considered the prevalence of tobacco smoking and betel-nut chewing to evaluate the impact of the nationwide oral mucosal screening program among men in Taiwan.

## 2. Methods

### 2.1. Data Source

This study used the database of all causes-of-death registration from 1991 to 2020 provided by the Ministry of Health and Welfare in Taiwan, and oral cancer deaths were extracted using the international classification of diseases codes 140, 141, 143–146, 148, and 149 from the ninth version from 1991–2007 and C00-C06, C09-C10, and C12-C14 from the tenth version from 2008–2020, respectively. Data on oral cancer death were categorized into sex, 5-year age groups, and single-year period groups. The number of people for the corresponding sex, 5-year age groups, and single-year period groups provided by the Department of Household Registration in the Minister of the Interior in Taiwan was used to calculate the mortality rate. This study excluded data for people younger than 29 years old because eligibility for the screening is restricted to people older than 30 years old. Data for those older than 65 were also excluded because the oral cancer mortality leveled off or decreased after that age. The age-specific mortality rates from oral cancer for those aged 30–64 are presented in Appendix A. The world (World Health Organization’s 2000–2025) standard population numbers were used to calculate the age-standardized mortality rates.

The prevalence of tobacco smoking among men in Taiwan was extracted from the online database of smoking prevalence in 1980–2015 from the Global Burden of Disease (GBD) Study provided by the Institute for Health Metrics and Evaluation, [17] which was categorized into 195 countries and territories by sex, and 5-year age groups. On the other hand, statistics on the prevalence of betel-nut chewing among men in Taiwan are lacking. This study used the databases of the National Survey on Knowledge, Attitude and Practice of Health Promotion in 2002, and the National Health Interview Survey in 2005, 2009, 2013, and 2017, respectively, and then used the annual consumption of betel nut (provided by the Council of Agriculture in Executive Yuan in Taiwan) as an independent variable to predict the annual prevalence of betel-nut chewing in a linear regression model. The log-odds transformation of betel-nut chewing prevalence was used for modeling. The prevalence of tobacco smoking and betel-nut chewing among men in Taiwan is shown in Appendix A.

### 2.2. Study Design

To evaluate the impact of the nationwide oral mucosal screening program on population health, this study estimated the expected oral cancer mortality if the screening program did not exist. However, this expected trend is counterfactual and unable to be directly estimated using the oral cancer mortality data. As a trade-off, this study used the risk fraction method to calculate the oral cancer mortality for subjects who did not smoke or chew before the implementation of the screening program (1991–1999). The linear trend (slope) of oral cancer mortality for these subjects was estimated using the ITS model and incorporated with the slope changes at the particular times (using the ITS model) by comparing them with female oral cancer mortality rates from 2000 to 2020 to analogize the oral cancer mortality for men from 1991 to 2020. (Because the prevalence of smoking or chewing is extremely low among women in Taiwan, the number of eligible people for the screening program among women is extremely small.) Then, the risk fraction method was used again to integrate all conditions of smoking or chewing together and calculate the expected trend of oral cancer mortality if the screening program did not exist. Therefore, the impact of the nationwide oral mucosal screening program among men in Taiwan could be evaluated by comparing the expected trend with the observed trend.

### 2.3. Risk Fraction

For a particular age and period group, let $P_{00}$ denote the prevalence of no tobacco smoking and no betel-nut chewing, $P_{01}$ denote the prevalence of chewing only, $P_{10}$ denote the prevalence of smoking only, and $P_{11}$ denote the prevalence of both smoking and chewing, where $P_{00}+P_{01}+P_{10}+P_{11}=1$. For the corresponding conditions of smoking or chewing, $r_{00}$, $r_{01}$, $r_{10},$ and $r_{01}$ are the oral cancer mortality rates, respectively. Let ${\mathrm{RR}}_{01}$, ${\mathrm{RR}}_{10},$ and ${\mathrm{RR}}_{11}$ denote the relative risks for smoking only, chewing only, and both of them, respectively, and the mortality rate for each exposed group is therefore:(1)$$r_{01}={\mathrm{RR}}_{01}\times r_{00}$$
(2)$$r_{10}={\mathrm{RR}}_{10}\times r_{00}$$
(3)$$r_{11}={\mathrm{RR}}_{11}\times r_{00}$$

These relative risks are assumed to be constant for all times and across different age groups. For all subjects (containing unexposed and exposed subjects together) at a particular age and period, the oral cancer mortality rate r can be directly calculated by the total number of oral cancer deaths and total number of people, and it is also a mixture of $r_{00}$, $r_{01}$, $r_{10},$ and $r_{01}$, that is,
(4)$$r=r_{00}\times P_{00}+r_{01}\times P_{01}+r_{10}\times P_{10}+r_{11}\times P_{11}.$$

Because $r_{00}$, $r_{01}$, $r_{10},$ and $r_{01}$ cannot be directly captured, Equation (4) is organized using Equations (1)–(3) to be:(5)$$r_{00}=\frac{r}{P_{00}+{\mathrm{RR}}_{01}\times P_{01}+{\mathrm{RR}}_{10}\times P_{10}+{\mathrm{RR}}_{11}\times P_{11}}$$
where $r_{00}$ can be calculated for given data of prevalence (90% of all chewers were also smokers among men in Taiwan according to previous studies), [8,18] relative risk, and mortality rate; therefore, $r_{01}$, $r_{10},$ and $r_{11}$ can also be calculated by Equations (1)–(3). Data for ${\mathrm{RR}}_{01}$, ${\mathrm{RR}}_{10},$ and ${\mathrm{RR}}_{11}$ were extracted from a cohort study which estimated the relative risks of oral cancer mortality among men in Taiwan for tobacco smoking or betel-nut chewing [18]. However, the ${\mathrm{RR}}_{01}$ (for subjects of chewing only) in this reference was not estimated because the death number from oral cancer was too rare to make a valid statistical inference. Under these circumstances, the method of relative excess risk due to interaction (RERI) was used and constrained to be zero to estimate the ${\mathrm{RR}}_{01}$. Here, RERI=0 indicates no relative excess risk due to interaction in an additive-effect relationship. Thus, ${\mathrm{RR}}_{01}=4.8$, ${\mathrm{RR}}_{10}=2.1,$ and ${\mathrm{RR}}_{11}=5.9$ were used in this study for risk fraction calculation.

To reasonably attribute the oral cancer deaths in a particular period to the extent of exposures in the same period or the past, Spearman rank correlation coefficients (ρ) between the oral cancer mortality (before the screening program) and the prevalence of exposures were calculated for a lag time of 0, 1, 3, 5, 7, …, and 15 years.

### 2.4. Interrupted Time-Series Analysis

The ITS model was constructed using the log-linear regression model with a simple linear spline function:(6)$$\log\left(r_{ij}\right)=\mu +\alpha_{i}+\beta_{1}T_{j}+\beta_{2}{\left(T_{j}-t_{1}\right)}_{+}+\beta_{3}{\left(T_{j}-t_{2}\right)}_{+}+\varepsilon_{ij},$$
where $r_{ij}$ denotes the oral cancer mortality rate at the *i*th age group (*i* = 1, 2, …, I) and *j*th period group (*j* = 1, 2, …, J), μ is the intercept, $\alpha_{i}$ is the age effect at the *i*th age group, $T_{j}=j$, ${\left(T_{j}-t_{1}\right)}_{+}=\max\left\{0,\left(T_{j}-t_{1}\right)\;\right\}$ and ${\left(T_{j}-t_{2}\right)}_{+}=\max\left\{0,\left(T_{j}-t_{2}\right)\;\right\}$ are the spline functions which use two knots of $t_{1}$ and $t_{2}$ as the time points of 1999 and 2007, respectively. Here, 1999 was the launch year of the screening program, and 2007 was the year the screening coverage rates began to rise (see Appendix A). The parameter $\beta_{1}$ refers to the linear slope of oral cancer mortality from 1991–1999, $\beta_{2}$ refers to the slope changes of oral cancer mortality from 1999 to 2007, and $\beta_{3}$ refers to the slope changes of oral cancer mortality after 2007. All data organization and statistical analysis were performed by using SAS software 9.4 version (SAS Institute Inc, Cary, NC, USA).

## 3. Results

Table 1 presents the Spearman’s ρ between oral cancer mortality rates among men in Taiwan from 1991 to 1999 and the prevalence of exposures (tobacco smoking and betel-nut chewing) for the different lag year. The Spearman’s ρ were 0.933 and 1 for tobacco smoking and betel-nut chewing in 11 lag years, respectively, and all were statistically significant with Bonferroni’s corrected alpha level (α = 0.0045).

Figure 1 presents the age-standardized mortality rates of oral cancer in Taiwanese men between the ages of 30 and 64 from 1991 to 2020. The age-standardized mortality rates increased from 10.14 deaths per 100,000 population in 1991 to 23.39 deaths per 100,000 population in 1999. The average annual percentage change of the age-standardized mortality rates was 10.9%. After the implementation of the screening program, the age-standardized mortality rate was 32.28 deaths per 100,000 population in 2007, and the average annual percentage change was 4.3% from 2000–2007. Since 2007, the age-standardized mortality rates have been leveling off. The age-standardized mortality rate was 33.49 deaths per 100,000 population in 2020, and the average annual percentage change was 0.4% from 2008 to 2020.

The oral cancer mortality trend among men with no tobacco smoking and no betel-nut chewing from 1991 to 1999 was calculated by Equation (5), and its linear slope (${\hat{\beta }}_{1}$) was 0.0783 (0.0581 to 0.0985). The parameter estimates ${\hat{\beta }}_{2}$ and ${\hat{\beta }}_{3}$ (slope changes) of ITS analysis were −0.0595 (−0.1109 to −0.0080) and 0.0069 (−0.0252 to 0.0389) for the oral cancer mortality rates among women (not shown in table or figure). Here, this study incorporated the linear slope in men (from 1991 to 1999) with the parameter estimates ${\hat{\beta }}_{2}$ and ${\hat{\beta }}_{3}$ for women (from 2000–2020) to analogize the oral cancer mortality rates among men with no tobacco smoking, no betel-nut chewing, and no influence of the screening program from 1991 to 2020 (${\hat{r}}_{00}$). Hence, Equations (1)–(3) were used to calculate the oral cancer mortality rates for men who smoked tobacco or chewed betel nuts, and then Equation (4) was used to combine these rates (${\hat{r}}_{00}$, ${\hat{r}}_{01}$, ${\hat{r}}_{10},$ and ${\hat{r}}_{01}$) to obtain the expected oral cancer mortality rates if there was no screening program. 

Figure 2 presents the observed age-specific mortality rates of oral cancer and the expected oral cancer mortality rates if there was not a screening program. The oral cancer mortality rates were all increasing for each age group from 1991 to 1999. If the screening program did not launch, the increasing trends of age-specific oral cancer mortality rates would be slightly weakened after 2000 and then changed to a leveling-off trend after 2009.

Figure 3 presents the observed and expected trends of age-standardized oral cancer mortality rate for Taiwanese men between 30 and 64 years old from 1999 to 2020. Overlapping trends between observed and expected age-standardized oral cancer mortality rates were found from 1999 to 2007. Thereafter, the expected trends (if the screening program did not launch) of age-standardized oral cancer mortality rates were leveling off but still higher than the observed trends.

Table 2 presents the effects (percentage changes in rates due to the screening program) of the nationwide oral mucosal screening program on oral cancer mortality. The percentage changes between the observed (implementation of the screening program) and expected (if the screening program did not launch) oral cancer mortality rate from 2000 to 2007 were not statistically significant after Bonferroni’s correction for each age group. For 2008–2020, observed and expected oral cancer mortality rates were statistically significant between 30 and 54 years old. The significant percentage changes in rates for age groups 30–34, 35–39, 40–44, 45–49, and 50–54 were −178.0% (99% CIs: −140.8 to −215.2), −75.4% (−59.4 to −91.4), −33.7% (−24.7 to −42.7), −18.8% (−12.0 to −25.6), and −15.3% (−9.5 to −21.2), respectively. The statistical reliability for the percentage changes in the 55–59 and 60–64 age groups are concerning because their bounds of confidence intervals of the point estimates were approaching null.

## 4. Discussion

One concern was the application of lag time. This study assessed the lag time of the temporal relationship between exposure to risk factors (tobacco smoking and betel-nut chewing) and oral cancer death according to the results of Spearman rank correlation coefficients. The lag time was used for risk fraction which involved the estimation of expected oral cancer mortality if there was no screening program. Inevitably, the ecological fallacy may confound the findings. However, the lag time of the highest Spearman’s ρ for both tobacco smoking and betel-nut chewing were the same (lag 11 years) which indicated that the temporal relationship of oral cancer death for each risk factor was consistent. In addition, previous studies also used 10 years or longer follow-up to assess the health risk of exposure for oral cancer, [19,20,21,22] and the reference which provided the relative risks of tobacco smoking and betel-nut chewing for oral cancer mortality among men in Taiwan was also a longitudinal study with 12 years of follow-up [18].

Using data among women to capture the slope changes was a consequence of a trade-off. So, the validity of using data from women rely on two study hypotheses. Firstly, the sex disparity of oral cancer mortality should be fully explained by the extent of exposure to the risk factors. According to the reports on high-mortality countries of oral cancer and their prevalence of risk factors [17,23,24]. oral cancer mortality among men was 1.2, 2, and 2.8 folds higher than those among women in Pakistan, India, and Sri Lanka, respectively, and their corresponding prevalence of risk factors among men were 3.8, 6.3, and 16.4 folds higher for tobacco smoking and 3.5, 2.3, and 2.5 folds higher for chewing tobacco (betel nut with smokeless tobacco) than those among women, respectively. As for the circumstance in Taiwan, the oral cancer mortality among men was 10-fold higher than among women, and the prevalence of tobacco smoking and betel-nut chewing (without smokeless tobacco) among men were 6.2 and 13.6 folds higher than among women, respectively. Although it is difficult to demonstrate that the sex disparity of oral cancer mortality could be fully explained by the extent of exposure to the risk factors, there is a tendency to observe a higher sex ratio of exposure prevalence in a country with a higher sex ratio of oral cancer mortality. Secondly, the impact of risk factors besides tobacco smoking and betel-nut chewing on the oral cancer mortality trend should be the same for men and women. In other words, the extent and trend of exposure prevalence of the other risk factors among men should be consistent with those among women. Excessive alcohol use is also a major risk factor for oral cancer mortality. In Taiwan, the prevalence of alcohol use was 0.5~0.6 among men and 0.2~0.35 among women in 2002–2017(the same data source with betel-nut prevalence and see Appendix A). Their sex ratio was nearly 2-fold for all times, and their prevalence trends were similar. Although the survey of alcohol use in Taiwan was approximate (lack of dose and frequency information), it still may explain the dramatically higher sex ratio in oral cancer mortality in Taiwan and also lead to estimating a lower oral cancer mortality trend and underestimating the effect of the oral mucosal screening program.

Furthermore, there is also a concern about using data among women. Although the prevalence of tobacco smoking and betel-nut chewing among women in Taiwan is very low, a few studies still indicate that nearly 14% of all screened subjects (from 2004 to 2009) were women [8,9,10]. Obviously, this statistic did not provide data relevant to the underlying trend of oral cancer mortality for abstinent subjects or those who had never undergone screening. Nevertheless, the impact of the screening program on oral cancer mortality among women was very little. The positive rates and detection rates among women were nearly 2 and 3 folds lower than those among men, and only 125 female oral cancer cases were found through the screening program from 2004 to 2009 which account for 4.5% of total female oral cancer cases [8,9,10]. Even though the influence of the screening program on women is real, it could have a consequence similar to the alcohol use prevalence, that is, to estimate a lower expected oral cancer mortality trend and underestimate the effect of the oral mucosal screening program. Under the circumstance of underestimation, this study still indicates the beneficial effect of the screening program on oral cancer mortality among men in Taiwan.

The risk fraction method used in this study is limited by the quality of the exposure prevalence, and it is also the reason that alcohol use was not considered in this study due to the low data quality. Moreover, only two risk factors (tobacco smoking and betel-nut chewing) and one interaction term were needed for statistical analysis in this study. While three or more risk factors are needed, it would be much more difficult to collect high-quality data. One possible solution is to use the causal-pie model proposed by Lee and his colleagues [25,26,27]. This method could take all the potential risk factors and their multiple interactions into account for estimating the weighted attributable fraction under a full causal pie, so that it could ensure a total of 100% for every attributed piece. A further study incorporating the quasi-experimental study design and sufficient component causal-pie model is warranted to evaluate more issues in public health and epidemiology.

In conclusion, this study incorporated the interrupted time-series analysis and risk fraction calculation to evaluate the nationwide oral mucosal screening program for oral cancer mortality among Taiwanese men and indicated that the nationwide oral mucosal screening may be beneficial to reduce oral cancer mortality rates in men aged 30 to 54, after considering the influence of tobacco smoking and betel-nut chewing. We found a lack of clear evidence to allay concerns of systematic or random errors in the study design. As mentioned before, competing risks and other factors may confound these results in older adults. Moreover, the validity of using data from women as the underlying trend may also affect these findings.

## Figures and Tables

**Figure 1 ijerph-19-14329-f001:**
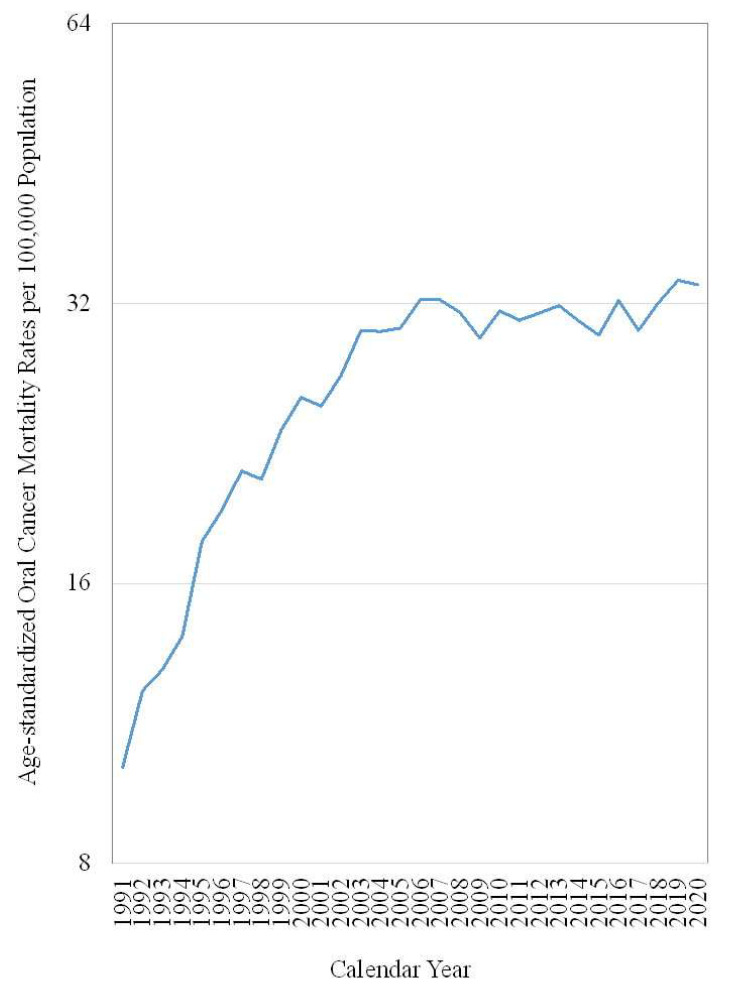
Age-standardized mortality rates of oral cancer in Taiwanses men between 30 and 64 years old from 1991 to 2020.

**Figure 2 ijerph-19-14329-f002:**
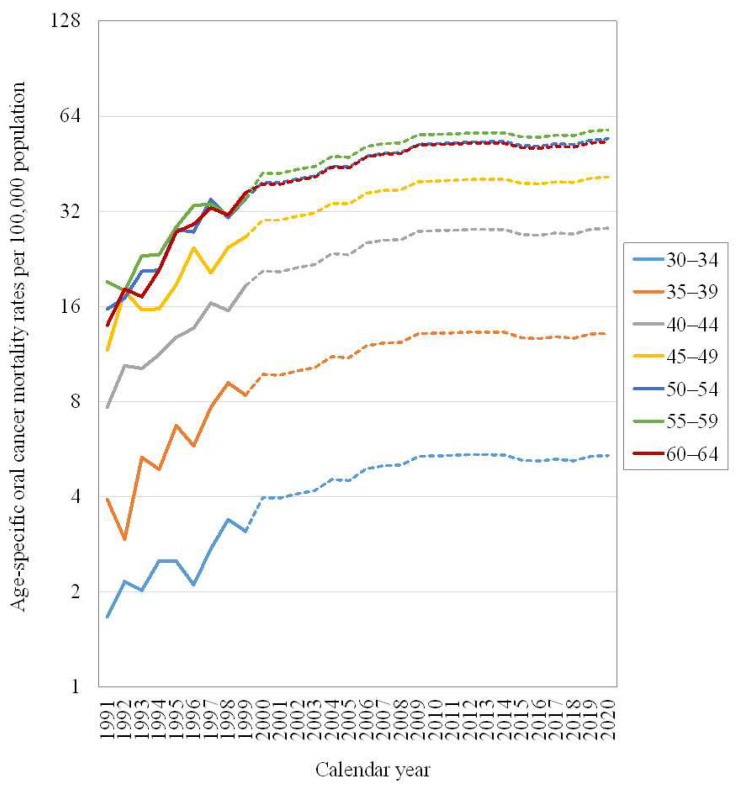
The observed (from 1991 to 1999) and expected (with no screening program in 2000–2020) age-specific mortality rates of oral cancer in men aged 30 to 64 in Taiwan.

**Figure 3 ijerph-19-14329-f003:**
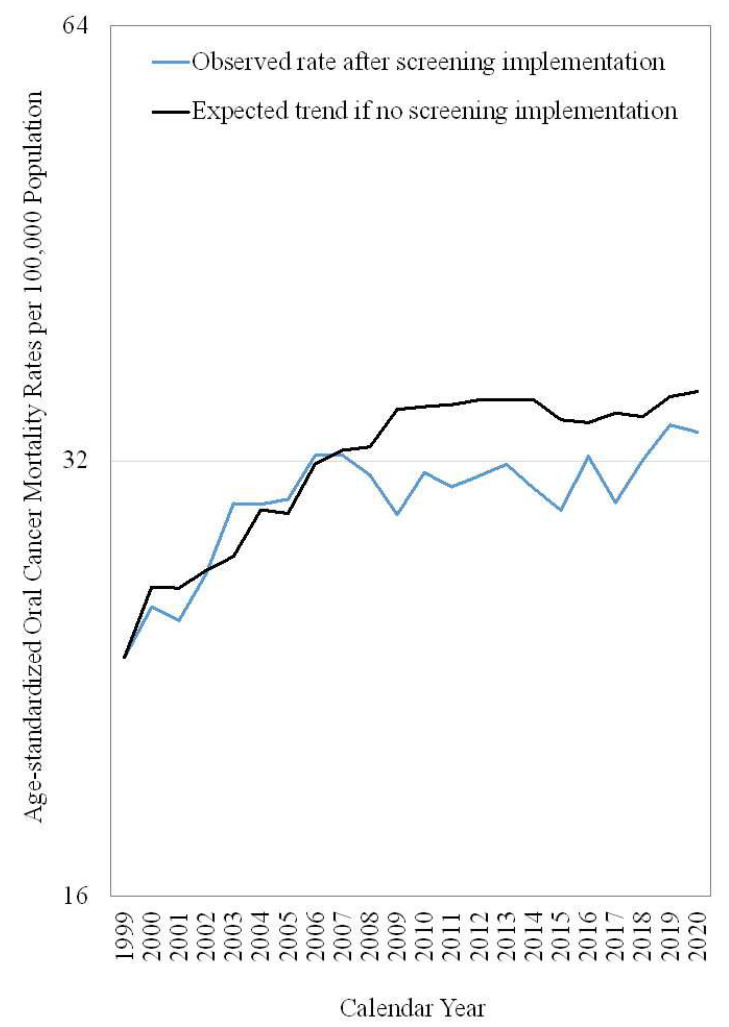
The observed and expected trends of age-standardized oral cancer mortality rate in Taiwanese men between 30 and 64 years old from 1999–2020.

**Table 1 ijerph-19-14329-t001:** Spearman rank correlation coefficients between oral cancer mortality among men in Taiwan from 1991 to 1999 and prevalence of exposures (tobacco smoking and betel-nut chewing) at various lag years.

Lag Year	Betel Nut Chewing		Tobacco Smoking	
	Spearman’s ρ	*p* Value	Spearman’s ρ	*p* Value
0	0.850	0.0037	0.650	0.0581
1	0.967	<0.0001	0.900	0.0009
3	0.933	0.0002	0.400	0.2861
5	0.917	0.0005	−0.650	0.0581
7	0.933	0.0002	−0.233	0.5457
9	0.933	0.0002	0.567	0.1116
11	1.000	<0.0001	0.933	0.0002
13	0.933	0.0002	not available	-
15	0.900	0.0009	not available	-

Lag 0 year indicates the correlation between mortality from 1991 to 1999 and prevalence of risk factor from 1991–1999. Lag 1 year indicates the correlation between mortality from 1991–1999 and prevalence of risk factor from 1990–1998.

**Table 2 ijerph-19-14329-t002:** Effects of the nationwide oral mucosal screening program on oral cancer mortality in Taiwanese men between 30 and 64 years old.

	Observed Trend after Screening Program		Expected Trend if Screening Program Does Not Launch		Percentage Changes of Mortality Rate Due to Screening Program (99% CIs) ^c^
	Deaths ^a^	Rates ^b^	Deaths ^a^	Rates ^b^	
between 2000 and 2007					
Age: 30–34	310	4.17	326	4.38	−5.2% (18.6 to −28.9)
Age: 35–39	812	10.55	826	10.73	−1.8% (12.8 to −16.3)
Age: 40–44	1628	21.25	1749	22.83	−7.4% (3.0 to −17.9)
Age: 45–49	2273	32.46	2318	33.10	−2.0% (6.7 to −10.7)
Age: 50–54	2370	42.00	2470	43.77	−4.2% (4.4 to −12.8)
Age: 55–59	1684	44.58	1776	47.01	−5.4% (4.8 to −15.6)
Age: 60–64	1429	46.69	1314	42.92	8.1% (18.8 to −2.6)
between 2008 and 2020					
Age: 30–34	233	1.91	648	5.31	−178.0% (−140.8 to −215.2)
Age: 35–39	917	7.40	1608	12.97	−75.4% (−59.4 to −91.4)
Age: 40–44	2478	20.61	3313	27.56	−33.7% (−24.7 to −42.7)
Age: 45–49	4005	33.51	4758	39.82	−18.8% (−12.0 to −25.6)
Age: 50–54	5311	45.46	6126	52.44	−15.3% (−9.5 to −21.2)
Age: 55–59	5597	52.70	5952	56.04	−6.3% (−0.7 to −11.9)
Age: 60–64	4638	55.54	4326	51.81	6.7% (12.7 to 0.8)

^a^ Indicates cumulative deaths; ^b^ indicates cumulative mortality rate per 100,000 person-year; ^c^ 99% confidence intervals were computed by the Wald method and using a z value of 2.92 (Bonferroni’s corrected α = 0.05/14).

## Data Availability

All data of this paper are available upon request to corresponding author.

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
