# Peer review of "Evaluation of Nationwide Oral Mucosal Screening Program for Oral Cancer Mortality among Men in Taiwan"

_ijerph, 2022, doi:10.3390/ijerph192114329_

Round 1
Reviewer 1 Report
The study assessed the prevalence trends of tobacco smoking and betel nut chewing and the impact of an oral cancer screening program on oral cancer mortality in Taiwan. They used interrupted time-series analysis. The results are interesting. The following is suggested for revision:
1) Introduction: The rationale for the study should be justified. The background should be concise and justify why the study is needed, what contribution it will contribute to the field. The second paragraph introduced the analytical method, which is not the objective of the study.
2) Methods: A concise introduction on the time-series approach and how it matches the study should be presented. The equations are necessary but please make it less mathematical. Currently it is hard to read.
3) Discussion: Please discuss the implications in light of the findings.
Author Response
Reviewer 1
The study assessed the prevalence trends of tobacco smoking and betel nut chewing and the impact of an oral cancer screening program on oral cancer mortality in Taiwan. They used interrupted time-series analysis. The results are interesting. The following is suggested for revision:
1) Introduction: The rationale for the study should be justified. The background should be concise and justify why the study is needed, what contribution it will contribute to the field. The second paragraph introduced the analytical method, which is not the objective of the study.
Author response:
Many thanks for your kindly comments and suggestions about this study. To enhance the research motivation, I have revised and added a few statements in the second paragraph of introduction section as below:
“Understanding the impact of the national screening program on the morbidity of target disease in the population could give important feedback to modify or improve the screening policy and strategy. However, it is difficult to evaluate the impact of the oral mucosal screening program on population health in Taiwan because of the selective screening strategy which involves the burden of tobacco smoking and betel nut chewing in the population. In other words, the expected screening effect (beneficial role) would be counteracted if the prevalence of smoking and chewing increased, and also would be ambiguous if the prevalence decreased. Previous studies using a database collected by the nationwide oral mucosal screening program in Taiwan evaluated the cost-effectiveness, reliability and validity, histopathology, and prognosis for oral cancer incidence and survival among eligible people who voluntarily receive screening and treatment.[8-10] How-ever, these studies were screening-based or uncontrolled designs, and cannot clarify the screening effect on population health. The previous study had also indicated a nearly leveling-off trend of oral cancer mortality in Taiwan since 2004,[11] but it could not clarify that this change in trend is due to the impact of the screening program or the decreasing prevalence of tobacco smoking and betel nut chewing. The exposed levels of risk factors (tobacco smoking and betel nut chewing) are strongly associated with oral cancer mortality, but there is no study that could take risk factors into account for evaluating the impact of the screening program on population health.”
2) Methods: A concise introduction on the time-series approach and how it matches the study should be presented. The equations are necessary but please make it less mathematical. Currently it is hard to read.
Author response:
To be honestly, it was not an easy task to conceptualize the study design, and write the paper with more readable, and also with less mathematical equations and formulas. The first draft paper had nearly 5000 words, and 28 equations and formulas. I had diminished a part of the mathematical section before submission. I understand that these mathematical statements would make it hard to read but they are crucial to keep the testability and reproducibility.
Although I cannot cut the method section, a statement about the process of statistical modeling and the parameter estimation has added (as a third paragraph in the results section) to improve readability as below:
“The oral cancer mortality trend among men for subjects with no tobacco smoking and no betel nut chewing from 1991-1999 was calculated by equation (5) and its linear slope () was 0.0783 (0.0581 to 0.0985). The parameter estimates and (slope changes) of ITS analysis were -0.0595 (-0.1109 to -0.0080) and 0.0069 (-0.0252 to 0.0389) from the oral cancer mortality rates among women (not shown in table or figure). Here, this study incorporated the linear slope in men (from 1991-1999) with the parameter estimates and in women (from 2000-2020) to analogize the oral cancer mortality rates among men for subjects with no tobacco smoking and no betel nut chewing and with no influence of the screening program from 1991-2020 (). Hence, equations (1), (2), and (3) were used to calculate the oral cancer mortality rates for each condition of smoking or chewing, and then equation (4) was used to combine these rates (, , and ) as the expected oral cancer mortality rates if screening program does not launch among men in Taiwan from 1991-2020.”
3) Discussion: Please discuss the implications in light of the findings.
Author response:
Due to the study limitation, I have added a conservatively conclusion statement about the implication of the study findings in the end of the discussion section as below:
“In conclusion, this study incorporated the interrupted time-series analysis and risk fraction calculation to evaluate the nationwide oral mucosal screening program for oral cancer mortality among Taiwanese men, and indicated that the nationwide oral mucosal screening may be beneficial to reduce the oral cancer mortality rates between 30 and 54 years old, after considering the influence of tobacco smoking and betel nut chewing prevalence on temporal changes. The effect of the screening program on oral cancer mortality rates between 55 and 64 years old among men was a lack of clear evidence to rid of the concerns of systematic or random errors from the study design. As mentioned before, the competing risk and other factors may confound these results in elders. Besides, the validity of using data from women as the underlying trend (oral cancer mortality rates for subjects with no tobacco smoking and no betel nut chewing and also with no influence of the screening program) may also affect these findings.”
Reviewer 2 Report
Dear author,
This is an interesting study evaluating the nationwide oral mucosal program for oral cancer mortality among men in Taiwan.
I have only minor comments:
- add a conclusion section
- revise the manuscript because there is minor spelling error and some spaces are missing.
Author Response
Reviewer 2
This is an interesting study evaluating the nationwide oral mucosal program for oral cancer mortality among men in Taiwan.
I have only minor comments:
- add a conclusion section
- revise the manuscript because there is minor spelling error and some spaces are missing.
Author response:
Many thanks for your comment. I have reviewed and revised the grammatical and spelling errors again to make this paper more readable.
In this revised version, I have added a conclusion statement in the end of discussion section as below:
“In conclusion, this study incorporated the interrupted time-series analysis and risk fraction calculation to evaluate the nationwide oral mucosal screening program for oral cancer mortality among Taiwanese men, and indicated that the nationwide oral mucosal screening may be beneficial to reduce the oral cancer mortality rates between 30 and 54 years old, after considering the influence of tobacco smoking and betel nut chewing prevalence on temporal changes. The effect of the screening program on oral cancer mortality rates between 55 and 64 years old among men was a lack of clear evidence to rid of the concerns of systematic or random errors from the study design. As mentioned before, the competing risk and other factors may confound these results in elders. Besides, the validity of using data from women as the underlying trend (oral cancer mortality rates for subjects with no tobacco smoking and no betel nut chewing and also with no influence of the screening program) may also affect these findings.”